# Gut Microbiota and Tumor Immune Escape: A New Perspective for Improving Tumor Immunotherapy

**DOI:** 10.3390/cancers14215317

**Published:** 2022-10-28

**Authors:** Yunbo He, Jinliang Huang, Qiaorong Li, Weiping Xia, Chunyu Zhang, Zhi Liu, Jiatong Xiao, Zhenglin Yi, Hao Deng, Zicheng Xiao, Jiao Hu, Huihuang Li, Xiongbing Zu, Chao Quan, Jinbo Chen

**Affiliations:** 1Department of Urology, Xiangya Hospital, Central South University, Changsha 410013, China; 2National Clinical Research Center for Geriatric Disorders, Xiangya Hospital, Central South University, Changsha 410013, China; 3Department of Ultrasound, Hunan Provincial People’s Hospital, The First Affiliated Hospital of Hunan Normal University, Changsha 410000, China; 4Department of Intensive Care Medicine, Xiangya Hospital, Central South University, Changsha 410013, China

**Keywords:** gut microbiota, tumor microenvironment, tumor immunotherapy, immune checkpoint inhibitor

## Abstract

**Simple Summary:**

The gut microbiota is a commensal microbiota living in the human intestine. Its status and composition have a profound impact on human antitumor immunity. Gut microbiota and its metabolites can influence tumor immune escape through immune cells and inflammatory factors, changing the patient’s response to immunotherapy. Protecting normal gut microbiota or optimizing its composition can improve the effects of tumor immunotherapy and bring new hope for cancer treatment.

**Abstract:**

The gut microbiota is a large symbiotic community of anaerobic and facultative aerobic bacteria inhabiting the human intestinal tract, and its activities significantly affect human health. Increasing evidence has suggested that the gut microbiome plays an important role in tumor-related immune regulation. In the tumor microenvironment (TME), the gut microbiome and its metabolites affect the differentiation and function of immune cells regulating the immune evasion of tumors. The gut microbiome can indirectly influence individual responses to various classical tumor immunotherapies, including immune checkpoint inhibitor therapy and adoptive immunotherapy. Microbial regulation through antibiotics, prebiotics, and fecal microbiota transplantation (FMT) optimize the composition of the gut microbiome, improving the efficacy of immunotherapy and bringing a new perspective and hope for tumor treatment.

## 1. Introduction

The gut microbiota comprises numerous anaerobic and facultative aerobic bacteria living in the human gastrointestinal tract [1]. A stable gut microbiota is indispensable for maintaining the normal physiological function of the human body, directly or indirectly involved in the body’s digestion, metabolism, immunity, and inflammation, etc. [2]. Abnormal changes in gut microbiota have been shown to induce or promote a variety of diseases and pathological conditions, including cancer [3,4,5,6]. With the in-depth study of the microbiota, researchers have proposed several ways through which gut microbiota affects distant organs, such as the Microbiota–Gut–Brain Axis [7], the Microbiota–Gut–Bone Axis [8], and the Microbiota–Gut–Liver Axis [9]. These findings have enriched our understanding of the mechanisms by which gut microbiota affects disease states. The important role of gut microbiota in the initiation and progression of cancer has also been gradually discovered. The gut microbiota and its metabolites were found to promote the occurrence and development of colon cancer and hepatocellular carcinoma (HCC) [10,11], and the gut microbiota may increase the risk of a poor prognosis for breast cancer patients by affecting estrogen metabolism [12].

The imbalance of gut microbiota is mainly the decrease of good bacteria and the increase of bad bacteria. Therefore, to change this dysregulation, increase the proportion of anti-tumor beneficial bacteria and restore the intestinal beneficial bacteria ecology, such as prebiotics and FMT [13,14], can fight against the occurrence and development of tumors. Diet therapy is also a highly concerned treatment. High cholesterol diet can induce dysbiosis of the microbiota, leading to the progression of hepatocellular carcinoma (HCC). Reducing dietary cholesterol intake in HCC patients may be beneficial to the treatment of HCC [15]. High dietary fiber diet is also beneficial to the gut microbiota ecology of cancer patients. Dietary fiber supplementation can improve the dysregulation of intestinal flora and improve the effect of tumor immunotherapy [16].

With the development of immune surveillance theory, tumor immune escape has been recognized as an indispensable mechanism of tumorigenesis [17,18]. Tumor immune escape refers to the process in which tumor cells evade the surveillance of the immune system, avoiding the clearance. Tumor immune escape is achieved through the interaction between three factors: the tumor microenvironment (TME), the tumor cells, and the immune status of the body [19,20,21]. Gut microbiota is closely related to the human immune system. It can affect tumor immune escape by its own structure and metabolites, which is an important reason for tumor progression and poor prognosis [22,23].

The goal of tumor immunotherapy is to restore antitumor immunity to control and eliminate tumors. Tumor immunotherapy includes cytokine therapy, adoptive immune cell therapy, and immune checkpoint inhibitor therapy, etc. [24]. Since gut microbiota affects the efficacy of tumor immunotherapy by influencing the function of immune cells and the release of cytokines and chemokines, microbiota regulatory strategies, such as fecal microbiota transplantation (FMT), antibiotic cocktails, and prebiotics, are potential adjuvants to tumor immunotherapy [25,26,27].

## 2. Gut Microbiota Influences Tumor Immune Escape through Tumor Microenvironment

The tumor microenvironment (TME) contains components that inhibit/promote the proliferation and invasion of tumor cells [28]. Treg cells, TAMs (tumor-associated macrophages), MDSCs (myeloid-derived suppressor cells), immunosuppressive molecules, and immunoregulatory enzymes promote tumor growth, proliferation, and metastasis, while immune effector cells and immune effector molecules inhibit them. In the TME, the immunosuppressive network composed of immunosuppressive cells and their secreted immunosuppressive molecules is the most important reason for tumor immune escape and resistance to antitumor therapy [29]. Gut microbiota can influence the status of key components in the TME, affecting tumor immune escape.

Inflammation plays an important role in tumor immune evasion triggered by gut microbiota. As early as 1863, Virchow et al. observed that malignancy tends to occur at sites of chronic inflammation [30], suggesting a correlation between inflammation and tumorigenesis. Subsequently, epidemiological data strongly suggested that chronic inflammation is associated with an increased risk of cancer [31,32]. At the same time, there is inflammation in TME, where infiltrating cells such as macrophages, neutrophils, monocytes, and mast cells together with the secretion of cytokines and the interaction with the tumor constitute the complex inflammation of the TME and are the foundation of tumor immune escape. Recent studies have reported that gut microbiota can invade colonic epithelial cells and activate the intracellular signaling system to trigger the host inflammatory response [33,34]. These inflammatory reactions are closely related to the occurrence of gastrointestinal tumors [35,36].

Particular attention should be paid to the fact that angiogenesis is a common feature of the inflammatory response and tumor progression. The inflammatory microenvironment promotes tumor development by accelerating angiogenesis and disrupting adaptive immune response [37,38]. It has long been noted that colonizing gut microbiota can induce intestinal angiogenesis in germ-free mice [39]. Later studies demonstrated that this angiogenesis is mediated through the tissue factor (TF) [40], a membrane receptor that initiates an exogenous coagulation pathway and can promote tumor angiogenesis [41,42]. Gut microbiota closely links inflammation to tumor progression by promoting angiogenesis. Metabolites of gut microbiota promote specific angiogenesis in a NOD-like receptor-dependent manner and are associated with chronic intestinal inflammation [43]. Mucosal *Escherichia coli* expressing the AFA-1 operon upregulates the expression of HIF-1α in the intestinal epithelial cells, promotes angiogenesis, and induces inflammation, leading to tumor progression [44,45,46].

Because of the temporal and spatial proximity, colorectal cancer (CRC) is currently considered to be the cancer most affected by gut microbiota that is closely related to biological dysregulation, extensive changes in gut microbiota, and enrichment of certain microbial strains [47,48]. Gut microbiota plays a role in maintaining the homeostasis of the normal intestinal environment. However, once the intestinal environment is changed under the induction of internal or external factors or the composition of gut microbiota is unbalanced, gut microbiota may aggravate the inflammatory reaction and release inflammatory factors. These inflammatory factors influence the TME, promoting the occurrence and development of CRC [49]. In addition, the microbiota located in the gut can also affect tumors outside the gut through the maturation and migration of the body’s own immune system, such as in leukemia [50,51,52] and lymphoma [53,54,55].

### 2.1. Gut Microbiota, Th17 Cells, and IL-17 Families

The accumulation of T helper 17 (Th17) cells in a variety of tumors promotes the occurrence and development of tumors [56,57]. The interleukin-(IL-)17 cytokine family consists of six members named IL-17A, IL-17B, IL-17C, IL-17D, IL-17E, and IL-17F. Th17 cells primarily produce IL-17A (also known as IL-17), IL-17F, and IL-22, cytokines that play an important role in inflammation [58]. It is estimated that almost two-thirds of primary sporadic CRCs have increased IL17 expression [59]. IL-17A levels are elevated in the stroma and intestinal epithelium of CRC in patients with colorectal adenoma throughout all stages of cancer [60]. Chronic IL-17 mucosal production may be the result of specific microorganisms or communities’ stimulation [61]. Segmented filamentous bacteria (SFB), a microbe of mice, has been found to adhere to the intestinal surface of mice, contributing to the differentiation of Th17 cells in the lamina propria of the mouse small intestine and inducing IL17 expression [62,63]. Bile acid metabolites have a similar effect, which may promote the malignant behavior of tumors. Bile acids undergo gut microbiota-mediated transformation in the intestine to produce many bioactive molecules, such as lithocholic acid (LCA) derivatives, which can reduce Th17 in the intestinal lamina propria and increase Treg cell differentiation [64].

IL-17 can induce and change the signaling pathway in colonic epithelial cells (CECs). Furthermore, it promotes the activation of T cells and stimulates epithelial cells, endothelial cells, and fibroblasts to produce a variety of cytokines, such as IL-6, IL-8, granulocyte-macrophage colony-stimulating factor (GM-CSF), and chemical activator and cellular adhesion molecule-1 (CAM-1). This leads to inflammation [65,66]. The present study confirmed a causal link between gut microbiota, IL-17-mediated chronic inflammation, and cancer [67,68,69]. Yang Z et al. reported that IL-17 upregulated the expression of CD39 and CD73 in Treg cells in the TME, enhancing the immunosuppressive function of Treg cells, and promoted the immunosuppressive ability of myeloid-derived suppressor cells (MDSCs) by upregulating IL-10 and IL-13, which promoted the immune escape of tumors [70] (Figure 1a). These observations support the conclusion that gut microbiota may regulate IL-17 production by participating in the differentiation of Th17 cells, promote the immunosuppressive function of MDSC and Treg cells, and, ultimately, lead to the immune evasion of CRC.

Th17 cells are also important mediators of gut microbiota affecting the TME of extramucosal tumors. Arianna et al. found that *Prevotella heparinolytica*, a commensal bacterium that promotes local and distant Th17 differentiation [71], may have a significant effect on the invasiveness of extramucosal tumors by affecting the differentiation of Th17 cells in the intestinal epithelium, independent of intestinal inflammation [72]. *Prevotella heparinolytica* promotes the differentiation of Th17 cells in the gut and their migration into the bone marrow (BM) of mice, where Th17 releases IL-17, contributing to the progression of multiple myeloma (MM). Multiple myeloma (MM) is a clonal malignancy of plasma cells characterized by the abnormal proliferation of clonal plasma cells in the bone marrow and overproduction of monoclonal antibodies [73]. IL-17 produced after *Prevotella heparinolytica* stimulation affects Th17 differentiation and migration, which induces STAT3 phosphorylation in the plasma cells of MM patients and promotes MM progression [72] (Figure 1b). Related to this, IL-17 was found to promote tumor growth through the IL-6–STAT3 signaling pathway [74]. A recent study has also reported that IL-17A can increase MM cell viability by upregulating Syk expression and activating the NF-κB signaling pathway [75].

In addition to IL-17A, other cytokines in the IL-17 family may also be involved in the gut microbiota-induced tumor progression. IL-17C can regulate the innate immune function of epithelial cells [76], and its upregulation promotes the progression of human CRC. Song et al. reported that changes in the gut microbiota caused the upregulation of IL-17C in intestinal epithelial cells and induced the expression of Bcl-2 and Bcl-xL, which ultimately supported the development of CRC [77]. IL-17RA belongs to the IL-17 receptor subfamily. IL-17A, IL-17B, IL-17C, and IL-17E (also known as IL-25) all activate IL-17 signaling through IL-17RA [78]. Therefore, further understanding of IL-17RA and other IL-17 receptor subfamily members can help us better understand the role of different cytokines in the gut microbiota–immune–cancer axis [79]. Targeting IL-17 and Th17 cells has yielded some results in the treatment of chronic inflammation [80]. We look forward to its success in tumor immunotherapy in the future.

### 2.2. Gut Microbiota and TAMs

The tumor-associated macrophage (TAM) is a main type of infiltrating immune cell in the TME [81]. TAMs are not a single cell population in phenotype and biological activity. In the TME, hypoxia, metabolic reprogramming, fibrosis, and various inflammatory factors can significantly alter the phenotype of TAMs [82,83,84,85,86]. Because of the heterogeneity and plasticity of TAMs, they may vary among different cancer types and even among different individuals of the same cancer [87]. The broad consensus is that macrophages can be divided into two types according to their functions: classically activated M1 or alternatively activated M2 phenotypes [88]. M1 macrophages target neoplastic cells and mediate antibody-dependent cellular cytotoxicity (ADCC) [89]. At the same time, M1 macrophages initiate an inflammatory response and promote antitumor immunity [90,91]. M2 macrophages can inhibit inflammation, induce angiogenesis, and promote tumor invasion and migration [92,93]. Unfortunately, during tumorigenesis, TAMs are more likely to be polarized into a protumor phenotype [94]. M2-polarized TAMs can inhibit cytotoxic T lymphocytes (CTL) [95], recruit Tregs [96], and release PGE2 and TGF-β [97] to change the TME to an immunosuppressive microenvironment [98], allowing tumor cells to escape immune surveillance (Figure 2b).

Gut microbiota can induce the polarization of macrophages to M1 or M2, affecting the process of many inflammatory diseases [99,100,101]. In the tumor microenvironment, the recruitment and differentiation of TAMs influenced by intestinal microbiota play a crucial role in evading the immune surveillance of tumor cells. High expression of cathepsin K (CTSK), a lysosomal cysteine protease, is associated with poor clinical prognosis in a variety of tumors [102,103,104]. Rui Li et al. reported that imbalance of gut microbiota, mainly the excessive proliferation of *Escherichia coli*, can promote the late progression and poor prognosis of CRC. Further studies revealed that as *Escherichia coli* increased in dominance, its release of lipopolysaccharide (LPS), a bacterial antigen, could upregulate CTSK secretion by CRC cells. In CRC mice, CTSK upregulation of CRC caused by intestinal microbiota imbalance can induce M2 polarization of TAMs, leading to rapid CRC progression [105]. *Enterotoxigenic Bacteroides fragilis* (ETBF) dysregulation can also induce M2 polarization of TAMs and promote immune escape of mouse CRC cells. In germ-free mice, ETBF colonization promotes the formation of a chronic inflammatory and immunosuppressive microenvironment by stimulating p-STAT3-mediated polarization of M2 macrophages, which in turn accelerates the progression of CRC [106]. Dysregulation of gut microbiota is closely related to M2 polarization of TAMs that contribute to tumor immune escape, and it is well-known that diet is a key determinant of gut microbiota homeostasis [107]. A high-fat diet (HFD) is an established risk factor for microbiota dysregulation and tumor progression [108]. In Apc^min/+^ mice, a HFD can induce the dysregulation of gut microbiota and reduce the level of short chain fatty acids (SCFAs), which mainly include acetate, propionate, and butyrate and are metabolites of gut microbiota [109]. Low levels of SCFAs activate the MCP-1/CCR2 axis, which promotes the recruitment and polarization of M2 TAMs and ultimately CRC progression [110] (Figure 2a). In another particular example, *Lactobacillus* metabolizes dietary tryptophan into indole. Activation of aryl hydrocarbon receptor (AhR) can promote TAMs from pancreatic ductal adenocarcinoma (PDAC) to immunosuppressive phenotype and inhibit the accumulation of CD8^+^ T cells in the tumor [111] (Figure 2a).

The role of TAMs in the development of tumor immune escape is so critical that researchers have increasingly focused on them as a target for cancer therapy [90,112]. Selective depletion of TAMs can increase CD8^+^ T-cell infiltration in the TME, stimulate antitumor immune response, and improve the efficacy of chemotherapy in PDAC and breast cancer mice in vivo [113,114]. A variety of drugs and therapies targeting TAMs have been developed, and remarkable results have been achieved in animal models [115,116,117,118]. Gut microbiota also has potential as a therapeutic approach for targeting TAMs. Supplementation of Apc^Min/+^ mice with *Akkermansia muciniphila (A. muciniphila)* inhibited the progression of CRC. This is caused by induction of TLR2/ NLRP3-mediated recruitment and polarization of M1 TAMs by *A. muciniphila* [119] (Figure 2c). Perhaps in the future, we can target TAMs as a powerful adjunct to antitumor therapy by regulating the gut microbiota through dietary therapy, FMT, antibiotic cocktails, or prebiotics. 

### 2.3. Gut Microbiota and Other Immune Cells in the TME

*Fusobacterium nucleatum* (*F. nucleatum*) is a component of the human oral and intestinal microbiota. Metagenomic analysis showed that *F. nucleatum* was abundant in CRC tissues and in metastatic lymph nodes [120]. *F. nucleatum* has been shown to promote CRC progression through multiple mechanisms, and a high abundance of *F. nucleatum* in human CRC tissues is associated with a low density of CD4^+^ T cells [120,121,122]. *F. nucleatum* was found to expand MDSCs, inhibit T-cell proliferation, and induce T-cell apoptosis in CRC [123]. Many studies have shown that *F. nucleatum* has immunosuppressive activity by inhibiting the response of human T cells to mitogens and antigens [124,125,126]. Tumors can use the Fap2 protein of *F. nucleatum* to inhibit the function of NK T cells through the T-cell immunoreceptor with immunoglobulin and ITIM domain (TIGIT), so that tumors can evade immune surveillance [127]. In addition, *F. nucleatum* also promotes M2 polarization of macrophages in *F. nucleatum*-related CRCs, which has an immunosuppressive effect [59] (Figure 1c).

In fact, the development of the mucosal immune system depends in part on innate immunity triggered by commensal microbes in the gut [128]. The colonization of *Bacteroides fragilis* (*B. fragilis*) and the production of polysaccharide A (PSA) in the mouse intestine promoted the development and differentiation of T cells [129]. At the same time, PSA recognized by toll-like receptors 2 (TLR2) on CD4^+^ T cells induces the development of Tregs [130]. Tregs can inhibit the function of tumor-specific T cells by secreting the inhibitory factors IL-10 and TGF-β [131] and activating the CTLA-4 pathway [132] to promote tumor immune escape. In CRC mice, the gut microbiota was found to downregulate CD8+ T-cell infiltration in the TME through activation of myeloid calcineurin, leading to tumor immune escape [133]. In addition, differences in T-cell differentiation caused by changes in gut microbiota can influence the effect of tumor chemoradiotherapy and regulate the response of patients to immunotherapy [134,135]. The role of gut microbiota in T-cell-centric immunotherapies such as CAR-T therapies and immune checkpoint inhibitors cannot be ignored. Elucidating the complex association and interaction between gut microbiota and immune cells, especially T cells, may be the key to resolving the nonresponse and side effects of immunotherapy in the future.

### 2.4. Gut Microbiota and PGE2

In the existing research, gut microbiota has been found to directly affect the immune evasion of intestinal tumors (mainly CRC) through inflammation and the tumor microenvironment. However, many studies have demonstrated that gut microbiota can affect gastrointestinal malignancies, but its driving mechanism in non-gastrointestinal malignancies remains to be elucidated. In addition to the gut microbiota mentioned above, which affects the differentiation of Th17 cells and leads to immune evasion of distant MM, gut microbiota can also affect the occurrence of nondigestive tract tumors through its metabolites.

Prostaglandin E2 (PGE2) is an important inflammatory factor in the TME, affecting the occurrence of tumors through a wide range of mechanisms and exerting immunosuppressive effects [136,137,138]. In HCC, gut microbiota can promote tumor development by increasing PGE2 in the TME. Metabolites of gut microbiota including lipoteichoic acid (LTA) and deoxycholic acid upregulate cyclooxygenase-2 (COX2) expression through toll-like receptor 2 (TLR2) on tumor cell membranes. At the same time, the expression of PGE2 increases and inhibits antitumor immunity under COX2 [139]. The activity of the COX2–PGE2 signaling pathway can inhibit dendritic cells (DCs), natural killer cells (NKs), and T cells, thereby promoting tumor immune escape [140]. Previous studies confirmed that COX2 inhibitors can inhibit tumor immune escape [138]. Therefore, gut microbiota may affect key immune cells in the TME through the TLR2–COX2–PGE2 signaling pathway and promote immune escape of HCC (Figure 3).

In addition to changes in the microbiota colonizing the gut affecting tumor immune escape through contact and noncontact modes, the microbiota colonizing other organs can also affect tumor progression. The sources of these organ microbiomes include gut microbiota migration and bacterial colonization from in vitro, which together with gut microbiota, play important roles in the process of tumor evasion from immune surveillance. In recent years, a variety of previously considered sterile tumors were found to have their own unique microbial composition, and tumor-resident intracellular microbiota has entered scientific view [141,142]. In pancreatic cancer, researchers have used antibiotic cocktail therapy and fecal microbiota transplantation (FMT) to demonstrate that the pancreatic cancer microbiome promotes tumor immune evasion by inducing innate and adaptive immunity [143]. The lung cancer microbiome promotes immunosuppression of lung cancer by inducing IL-17A production by γδT cells [144]. The tumor microbiome is increasingly recognized as an indispensable part of the TME, which may serve as a bridge between gut microbiota and tumor immune evasion and is expected to become a new target for cancer treatment. These findings suggest a complex relationship between gut microbiota, inflammation, and the TME. Their interactions constitute an immunosuppressive network that links gut microbiota with tumor immune escape. Understanding the relationship between microbes, immune cells, and inflammatory factors in the TME will provide a new perspective and method for tumor immunotherapy.

## 3. Gut Microbiota and Tumor Immunotherapy

The proliferation of tumor cells is the result of the tumor evading the surveillance of the host immune system through a variety of mechanisms, such as the induction of immunosuppression in the TME and the downregulation of the expression of target antigens [145]. In recent years, the enhanced understanding of the tumor immune escape mechanism has led to the rapid development of tumor immunotherapy [146]. Tumor immunotherapy can help the body rebuild or strengthen antitumor immunity through various ways to cure or delay the tumor process [147,148]. Tumor immunotherapy includes chimeric antigen receptor (CAR)-T-cell therapy, immune checkpoint inhibitors (ICIs), CpG oligonucleotides, etc. Gut microbiota has been found to influence the response to immunotherapy and the occurrence of complications by modulating the infiltration degree of immune cells and the secretion of inflammatory factors in the TME [149,150,151]. We believe that further elucidation of the involvement of gut microbiota in the mechanism of tumor immune evasion will guide the development of tumor immunotherapy.

### 3.1. Gut Microbiota Affects the Efficacy of Immune Checkpoint Inhibitors

ICIs target the regulatory pathways of T cells and re-expose tumor cells that have escaped immune surveillance to enhance antitumor immune responses [152]. Antibodies that block cytotoxic T lymphocyte-associated protein 4 (CTLA-4) or programmed cell death 1 (PD-1) pathways are the most commonly used ICIs [153,154]. ICIs have been shown to have good efficacy in melanoma, nonsmall cell lung cancer, and other tumors, but the interpatient variability of the treatment effect is still a concern [155].

In an ICI treatment targeting the PD-1/PD-L1 axis in melanoma patients, Response Evaluation Criteria in Solid Tumors (RECIST) were used to distinguish melanoma patients with or without a clinical response to ICI treatment, and they were labeled as responders or nonresponders. There was a significant difference in fecal microbial composition between responders and nonresponders. When fecal microbiota from responders and nonresponders were transplanted into melanoma mice that received PD-1 blockade treatment, the response was significantly enhanced in mice that underwent responder feces, while the response was not observed in mice that underwent nonresponders [156]. This suggests that the gut microbiota includes both favorable and unfavorable microbiota, and the composition of gut microbiota plays a role in regulating the therapeutic effect. Analysis of the TME revealed that this effect was due to enhanced activation of tumor antigen-specific CD8^+^ T cells [157,158]. Higher concentrations of CD8^+^ T cells were clustered in the TME of mice receiving fecal microbial transplantation from “responders”, and the infiltration degree of CD8^+^ T bacteria was positively correlated with *Faecalibacterium* [159]. *Bifidobacterium* can directly induce DC maturation and cytokine synthesis and increase the number of CD8^+^ T cells in peripheral areas and in the TME [160]. In addition to CD8^+^ T cells, favorable flora can also increase the number and function of CD4^+^ T cells in the body. An abundance of *Faecalibacterium* cells can improve the number of circulating CD4^+^ T cells and promote cytokine response [159]. In view of the effects of gut microbiota on T cells, artificial regulation of microbiota composition is an effective strategy to assist ICI. Oral administration of *Akkermansia muciniphila* could induce the secretion of IL-12 from DCs and increase the accumulation of CCR9^+^CXCR3^+^CD4^+^ T lymphocytes in mouse tumors, thus recovering the efficacy of the PD-1 blockade [161]. The mechanism of gut microbiota influencing the CTLA-4 blockade effect is similar to that of the PD-1 blockade. *Bacteroides* species play an important role in it. Fecal microbial transplantation from humans to mice confirmed that *B. fragilis* and *B. thetaiotaomicron* promote tumor CTLA-4 blockade reactivity by promoting the IL-12-dependent TH1 immune response [162].

The mechanism by which favorable and unfavorable microbiota can regulate the efficacy of ICI in the intestinal tract has not been clearly reported, which may be related to the metabolic function of the microbiota. Favorable microbiota can synthesize a variety of metabolites that promote host immune function, while the catabolic function of unfavorable microbiota has the opposite effect [163]. In CRC model mice, inosine produced by the metabolism of *B. pseudolongum* was confirmed to enhance antitumor immunity induced by anti-CTLA-4 treatment, mediated by T cells [164]. In another study in mice, short chain fatty acids (SCFAs) produced by gut microbiota inhibited CD80/CD86 upregulation on dendritic cells induced by anti-CTLA-4 treatment, limiting the efficacy of anti-CTLA-4 in mice with metastatic melanoma [165]. The roles of favorable and unfavorable microbiota are not static. For example, in MSS-type CRC tumor-bearing mice, abundant *Bacteroides* would lead to increased Tregs and MDSCs in circulation and decreased CD8^+^ T-cell infiltration in the tumor bed, which would decrease the cytokine response and the therapeutic effect of the PD-1 blockade [166]. *Bacteroides* has the role of promoting the therapeutic effect in the CTLA-4 blockade mentioned above [162], and the reason for the difference between the two remains to be discussed.

### 3.2. Gut Microbiota and Antibiotic Use in ICI Therapy

In cancer patients, antibiotics are commonly used to prevent and treat a range of potentially life-threatening bacterial infections, which complicates cancer treatment and worsens the prognosis of patients with tumors [167,168,169]. It is a clinically obvious phenomenon that the long-term application of broad-spectrum antibiotics can lead to the irreversible destruction of gut microbiota ecology, which might result in diarrhea, double infection, and even the selection of drug-resistant strains [170,171,172]. In addition to increasing the complexity and difficulty of tumor treatment, dysregulated gut microbiota can also cause resistance to chemotherapy and immunotherapy in cancer patients [173,174]. A meta-analysis of 23 eligible studies showed that antibiotic use before or during ICI treatment resulted in a reduction in median OS for more than six months [175]. A retrospective study suggested that early use of antibiotics may be more harmful, while use of antibiotics during ICI treatment may be safer [176].

At present, it seems reasonable to avoid long-term and broad-spectrum antibiotics before starting ICI treatment, if possible. However, for some patients with malignant tumors, the use of antibiotics to prevent and treat infection is an irreplaceable choice because of long-term consumption and the decline of immunity [177,178]. For patients who do require antibiotics prior to ICI treatment, favorable strategies may be receiving FMT [157,179] or probiotics combined with dietary fiber therapy [180] in an attempt to eliminate the possible harmful effects of broad-spectrum antibiotics in ICI treatment. Another valuable direction would be to reduce the concentration of antibiotics in the gut without affecting their plasma pharmacokinetics. For example, nonspecific adsorbents can be applied to the patient’s intestine [181,182,183]. Oral β-lactamases target β-lactam antibiotics [184,185,186], significantly reducing the damaging effects of antibiotics on gut microbiota.

### 3.3. Gut Microbiota Affects the Efficacy of CAR T-Cell Therapy

Chimeric antigen receptor T (CAR-T) cell therapy refers to the modification of immunoreactive T cells derived from tumor patients with the chimeric antigen receptor (CAR), in vitro expansion, and then transfusion back into patients. The aim is to establish a long-term specific antitumor immune effect [187]. Therapeutic efficacy is usually limited by treatment-related toxicity, T-cell dysfunction, the TME, tumor antigen loss, and other reasons [188]. Antibiotic use is closely related to CAR-T efficacy, which is mediated by gut microbiota. In mice treated with broad-spectrum antibiotics, the tumor-suppressive effect of CAR-T was decreased [158]. Vancomycin-treated mice were more responsive than untreated mice to CAR-T treatment because vancomycin-screened gut microbiota induced an increase in systemic CD8α + DC and maintained adoptively transferred antitumor T cells in an IL-12-dependent manner [134].

The efficacy of CAR-T is affected by lymphocyte infiltration in the TME and the function of effector molecules. Vancomycin treatment increased the levels of granzyme B, perforin 1, IL-12, and IFN-γ in the TME of mice, which are all antitumor effector molecules [134]. The expression of antitumor effector molecules is affected by the metabolites of gut microbiota. Butyrate affects the gene expression of effector molecules by inhibiting histone deacetylase (HDAC) in CD8^+^ T cells. Higher concentrations of acetate can promote IFN-γ production by CD8^+^ T cells by regulating cell metabolism and mTOR activity, both of which affect the efficacy of CAR-T in mice [189]. In addition to effector molecules, gut microbiota can also modulate tumor-infiltrating lymphocytes (TILs). In human CRC, the gut microbiota stimulates tumor cells to produce chemokines to recruit antitumor T cells into the tumor tissue. Among them, the abundance of *Rikenellaceae*, *Ruminococcace,* and *Lachnospiracee* is significantly correlated with the expression levels of chemokines CCL5, CCL20, and CXCL11 [190].

A better understanding of the mechanisms by which the gut microbiome regulates the effects of CAR-T and ICI therapies will help maximize the effectiveness of immunotherapies in the future and reduce immunotherapy-related side effects [191]. A serious side effect of CTLA-4 blockade is autoimmunity, inducing severe inflammation. An experiment on mice showed that *Bifidobacterium* can relieve the toxicity of CTLA-4 blockade by affecting the metabolic function of Treg cells without affecting antitumor immunity [192]. Many immunotherapy strategies have been practiced and validated in experiments or clinical practice. More attention has been paid to the relationship between gut microbiota and antitumor immunotherapy. However, the role of gut microbiota in other immunotherapies, such as cytokine therapy and immune vaccines, remains poorly investigated. We believe that the regulation of gut microbiota will become an effective and necessary means of adjuvant antitumor immunotherapy with the further elucidation of the influence of gut microbiota on tumor immune escape.

## 4. Conclusions and Future Directions

Tumor immune escape is the process of tumor cells escaping from monitoring and killing by the immune system, while antitumor immunotherapy is the process of assisting the immune system to monitor and eliminate tumor cells. There is a complex crosstalk between gut microbiota, inflammation, and the TME. The colonization and metabolites of gut microbiota affect tumor escape from immune surveillance through complex mechanisms and the effect and complications of antitumor immunotherapy. The mechanism by which gut microbiota helps tumor cells escape immune surveillance has been identified, but many questions remain unanswered. Tumor cells themselves can escape the attack of tumor killer cells through a variety of ways, such as immune editing, low expression of MHC class I molecules, abnormal costimulatory signals (decreased expression of CD86 and upregulated expression of PD-L1), and expression or secretion of immunosuppressive factors TGF-β and IL-10 [193]. Studies have confirmed that gut microbiota can promote or inhibit tumor progression by activating intracellular signaling pathways through metabolites [22,194,195]. This suggests that gut microbiota may have a mechanism that directly enables tumor cells to evade immune surveillance. However, current studies on the regulation of antitumor immunity by gut microbiota have mainly focused on the regulation of immune cells and inflammatory factors outside tumor cells. There is no report of gut microbiota causing changes in tumor cells to escape immune surveillance. We look forward to further research revealing the status and role of gut microbiota in the network of tumor immune evasion.

The metabolites of gut microbiota are important mediators in the regulation of antitumor immunity. In the process of exploring the relationship between gut microbiota and tumor immunity, researchers have found a variety of metabolites that can affect tumor progression and antitumor immunity. These metabolites can still play an antitumor immune role or assist in enhancing the effect of immunotherapy in the absence of gut microbiota [164,196]. Elucidating the mechanism by which these metabolites act as antitumor drugs and developing targeted drugs or metabolite analogues are the two foreseeable industrial transformation directions regarding gut microbiota. At the same time, the development of inhibitors or antagonists [197] targeting tumor-promoting gut microbial metabolites may also be a promising prospect for adjuvant tumor immunotherapy.

The composition of gut microbiota plays an important role in tumor progression. Therefore, manipulating gut microbiota may be a new approach to improve tumor immunotherapy. At present, some microbiota interventions, such as FMT, prebiotics, probiotics, and antibiotic cocktail therapy, have been developed. Manipulation can be used to study the gut microbiota or as an adjunct to the experimental treatment of tumors. These strategies have shown promise as modulators of the gut microbiome. In general, gut microbiota can affect the process of tumor immune escape and the efficacy of tumor immunotherapy. Targeting gut microbiota and its metabolites to detect, block, adjust, or transplant is important for tumor immunotherapy intervention and prognosis assessment.

## Figures and Tables

**Figure 1 cancers-14-05317-f001:**
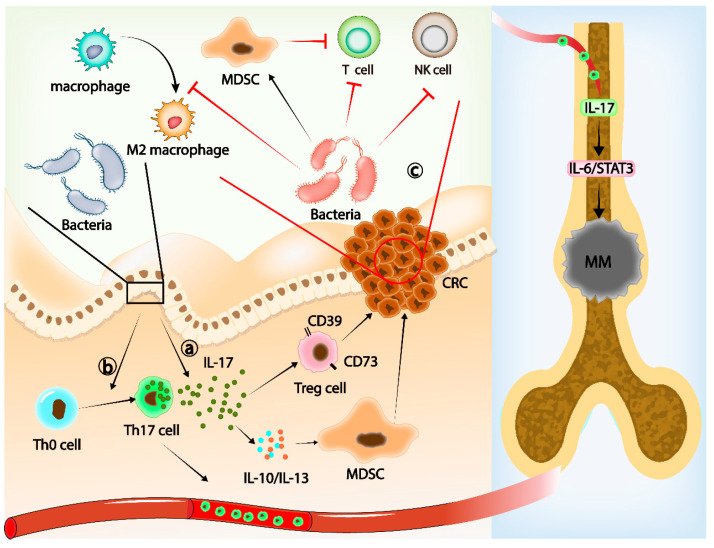
Gut microbiota, Th17 cells, and IL-17 families. (**a**) The increase in IL-17 induced by gut microbiota can upregulate the expression of CD39 and CD73 in Treg cells and enhance the immunosuppressive ability of Treg cells. Meanwhile, IL-17 promotes the immunosuppressive ability of MDCS by upregulating IL-10 and IL-13, thus promoting tumor immune evasion. (**b**) Gut microbiota induces Th0 cells to differentiate into Th17 cells and migrate into the bone marrow. Il-17 released from Th17 cells in the bone marrow promotes MM progression through the IL-6–STAT3 axis. (**c**) In CRC, gut microbiota promotes the function of MDSCs, induces T-cell apoptosis, inhibits NK cell function, and promotes M2 polarization of macrophages.

**Figure 2 cancers-14-05317-f002:**
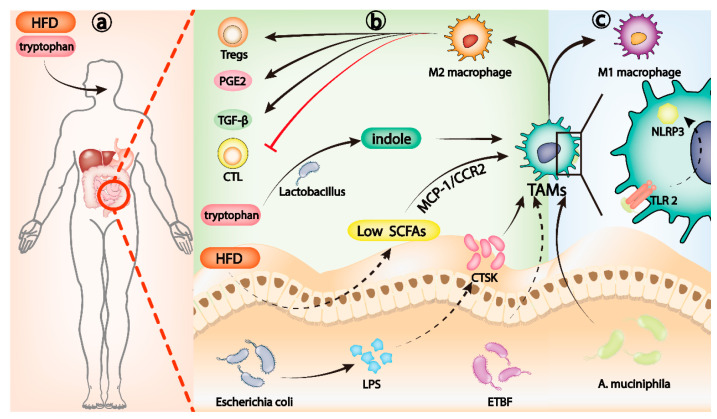
Gut microbiota, Th17 cells, and IL-17 families. (**a**) HFD can induce the dysregulation of gut microbiota and reduce the level of SCFAs. Low levels of SCFAs activate the MCP-1/CCR2 axis, which promotes the recruitment and polarization of M2 TAMs. Lactobacillus metabolizes dietary tryptophan into indole, which can activate aryl hydrocarbon receptor (AhR) to transform TAMs to immunosuppressive phenotype. (**b**) LPS released by Escherichia coli can upregulate CTSK secretion and induce M2 polarization of TAMs. Dysregulation of ETBF can also induce M2 polarization of TAMs. M2-polarized TAMs can inhibit cytotoxic T lymphocytes (CTL), recruit Tregs, and release PGE2 and TGF-β to change the TME to an immunosuppressive microenvironment. (**c**) *A. muciniphila* induces TLR2/NLRP3-mediated recruitment and polarization of M1 TAMs.

**Figure 3 cancers-14-05317-f003:**
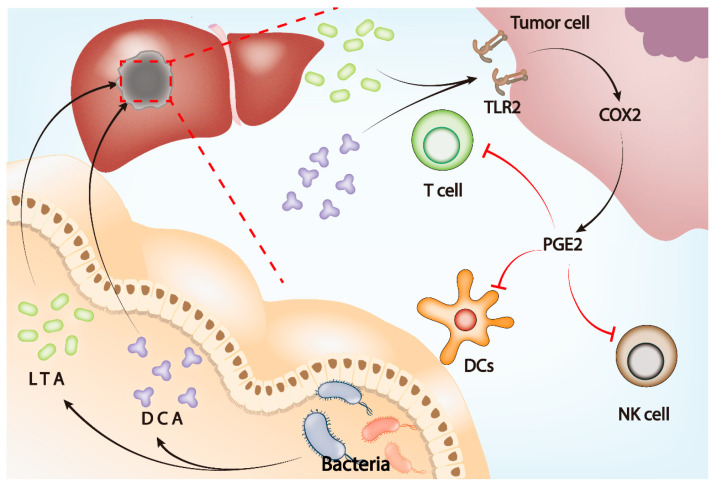
Gut microbiota influences tumor microenvironment through the gut–liver axis. LTA and DCA, metabolites of gut microbiota, upregulate the expression of COX2 by activating TLR2 on the tumor cell membrane, leading to the upregulation of PGE2 in the tumor microenvironment. Elevated PGE2 inhibits the function of DCs, T cells, and NK cells and ultimately induces an immunosuppressive microenvironment to assist tumor immune escape.

## Data Availability

Not applicable.

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
