# Peer review of "Gut Microbiota and Tumor Immune Escape: A New Perspective for Improving Tumor Immunotherapy"

_cancers, 2022, doi:10.3390/cancers14215317_

Round 1

Reviewer 1 Report

Tumor immune escape is the process of tumor cells escaping from the monitoring and killing of the immune system, while antitumor immunotherapy is the process of assisting the immune system to monitor and eliminate tumor cells. Stable gut microbiota is directly or indirectly involved in the bodyʹs immunity and inflammation. There is a complex crosstalk between gut microbiota, inflammation and the tumor microenvironment(TME). The colonization and metabolites of gut microbiota affect tumor escape from immune surveillance through complex mechanisms, and affect the effect and complications of anti-tumor immunotherapy. Studies have confirmed that gut microbiota can promote or inhibit tumor progression by activating intracellular signaling pathways through metabolites. This suggests that gut microbiota may have a mechanism that directly affects tumor cells to evade immune surveillance. There is no report that gut microbiota causes changes in tumor cells to escape immune surveillance. The metabolites of gut microbiota are important mediators in the regulation of anti-tumor immunity. The development of inhibitors or antagonists targeting tumor-promoting gut microbial metabolites may also be a promising prospect for adjuvant tumor immunotherapy. Therefore, manipulating gut microbiota may be a new approach to improve tumor immunotherapy. In this paper, the authors review the mechanisms by which gut bacteria may play a role in tumor immune escape. It is relatively innovative and interest and it is also rare in the published reviews. The latest research progress in all relevant research fields has been summarized and elaborated in details. In total, I recommend that this paper deserves publication in this journal.The manuscript needs to be carefully corrected for some incorrect formatting and improved readability.

Author Response

On behalf of my co-authors, we are very grateful to you for giving us an opportunity to revise our manuscript. We reviewed the effect of gut microbiota on tumor immune escape and its guiding role in tumor immunotherapy. We hope that these works will benefit the study of gut microbiota and help the development of tumor immunotherapy We appreciate it very much for your good suggestion, and we have studied your comments carefully and tried our best to revise our manuscript according to the comments. We rechecked and corrected the formatting errors to meet the requirements of MDPI. This article has been edited in English by MDPI(ID: english-51858) to improve readability.

Reviewer 2 Report

This review addresses an interesting and actual topic. However it was really hard and unpleasant to read, besides the need for a deep English language editing. The structure of the review is confused, the same division in chapters is overlapping: the sesction on PGE should have been included in that about TME, all the other chapters are somehow about T cells... why a chapter T cells, one for Th17, one for ICI, one for CAR-T?

The whole organization of the paper seems illogical, with overlap of arguments in different parts. Furthermore, the results of resesarch on human beings and on experimental animals are presented rather confusely, often not specifyng the nature of the descsribed researches.

As it is, the paper is more confusing than informing.

Author Response

       We are very sorry that our paper does not provide readers with readable enough information. In response to your suggestion, we have made the following changes:

  1. Readability: We have sought the help of the professional english editing service of MDPI to conduct an in-depth English editing of our paper. (ID: english-51858)
  2. Paper structure: We reorganized the structure of the paper, and revised the title and main body. The first part of the text mainly describes the influence of gut microbiota on different components of the TME, which leads to immune escape of tumor. We described the effects of gut microbiota on different components of the tumor microenvironment leading to tumor immune escape (immune cells: Th17, TAMs and other immune cells, inflammatory cytokines: PGE2). Here, weadd the literature review on the influence of gut microbiota on tumor immune escape through TAMs. (2.2 Gut microbiota and TAMs) The second part reviews the influence of the existing gut microbiota on the effect of immunotherapy. In the main text, we propose that further elucidation of the effect of gut microbiota on the mechanism of tumor immune escape can guide the development of tumor immunotherapy.(line 379-393)
  3. Nature of the study: Sorry that we did not strictly distinguish between the results of human and animal experiments, We now explain in the main text the sources of the results of each experiment.

Reviewer 3 Report

The authors have beautifully explained the role of GUT microbiome in the tumor progression and its relationship with the tumor evasion. This article provides a better understanding of the role of the microbiome composition and the effect in tumor and its microenvironment.

Following are the concerns for the article:

1.       The overall writing is divided in two parts, first part focuses mostly on the role of microbiome composition on tumor resistance and evasion, second half discusses mostly on the positive effect of the microbiome on the immunotherapies. This seems little contradictory. Please provide more of literature search for the first part (headings 1-5) for microbiome and its effect in anti-tumor effect and mechanism for it as well.

2.       Wherever possible please elaborate on the microbiome composition on itself as it would help readers to get better picture of the context.

3.       Line 83: Please clarify the statement for the inhibit/promote the proliferation and invasion of tumor cells. Provide the components of TME in detail.

4.       Line 87: Provide more context of contact and non-contact pathways and its relationship with GUT microbiota. Also, please add reference to this section.

5.       Line 89-102: Angiogenesis is also once of the factor mediated by inflammation. Please include the role of GUT microbiota in angiogenesis and its relation to cancer progression.

6.       Figure 1: Figure labels seems to be different in figure and legends: Please make sure a, b and c in caption matches the information in figure.

7.       Line 260- “in vitro bacterial colonization’—Should it be in vivo? Please clarify.

8.       Heading 6-8— For the cancer immunotherapy please include more examples where microbiome acts against the immune therapy. This would provide unbiased examples and proper information for the readers. Also, please provide more information on what type of cancer and treatments are discussed.

9.       Line 300- Please provide the context of responders and non-responders.

Author Response

Point 1: The overall writing is divided in two parts, first part focuses mostly on the role of microbiome composition on tumor resistance and evasion, second half discusses mostly on the positive effect of the microbiome on the immunotherapies. This seems little contradictory. Please provide more of literature search for the first part (headings 1-5) for microbiome and its effect in anti-tumor effect and mechanism for it as well.

Response:We reorganized the structure of the article and made changes to the title and body. The first part mainly describes the influence of intestinal flora on different components of TME, which leads to immune escape of tumor. We also add the literature review on the influence of gut microbiota on tumor immune escape through TAMs. The second part reviews the influence of the existing gut microbiota on the effect of immunotherapy. In the main text, we propose that further elucidation of the effect of gut microbiota on the mechanism of tumor immune escape can guide the development of tumor immunotherapy. We also provide an additional literature search on the microbiome and its antitumor effects and mechanisms. (refs. 13-16,119, etc.)

Point 2: Wherever possible please elaborate on the microbiome composition on itself as it would help readers to get better picture of the context.

Response:We appreciate it very much for this good suggestion, and we have done it according to your ideas. In the latest revision, we have explained the composition of the microbiome in as much detail as possible.

Point 3: Line 83: Please clarify the statement for the inhibit/promote the proliferation and invasion of tumor cells. Provide the components of TME in detail.

Response:We appreciate it very much for this good suggestion, and we have done it according to your ideas. In the revised manuscript, lines 92 to 97, we clarify statements regarding inhibition/promotion of tumor cell proliferation and invasion. The components of TME are also provided in detail.

The tumor microenvironment (TME) contains components that inhibit/promote the proliferation and invasion of tumor cells [28]. Treg cells, TAMs (tumor-associated macrophages), MDSCs (myeloid-derived suppressor cells), immunosuppressive molecules, and immunoregulatory enzymes promote tumor growth, proliferation, and metastasis, while immune effector cells and immune effector molecules inhibit them.

Point 4: Line 87: Provide more context of contact and non-contact pathways and its relationship with GUT microbiota. Also, please add reference to this section.

Response: We appreciate for this good suggestion. We consider the original expression to be inappropriate and have removed it. Because classifying the pathways of gut microbiota in contact and non-contact ways here would affect the structure of the text.

Point 5: Line 89-102: Angiogenesis is also once of the factor mediated by inflammation. Please include the role of GUT microbiota in angiogenesis and its relation to cancer progression.

Response: We appreciate it very much for this good suggestion, and we have done it according to your ideas. In the revised manuscript, lines 116 to 130, We added the role of gut microbiota in angiogenesis and its relationship to cancer progression.

Particular attention should be paid to the fact that angiogenesis is a common feature of the inflammatory response and tumor progression. The inflammatory microenvironment promotes tumor development by accelerating angiogenesis and disrupting adaptive immune response [37,38]. It has long been noted that colonizing gut microbiota can induce intestinal angiogenesis in germ-free mice [39]. Later studies demonstrated that this angiogenesis is mediated through the tissue factor (TF) [40], a membrane receptor that initiates an exogenous coagulation pathway and can promote tumor angiogenesis [41,42]. Gut microbiota closely links inflammation to tumor progression by promoting angiogenesis. Metabolites of gut microbiota promote specific angiogenesis in NOD-like receptor-dependent manner and are associated with chronic intestinal inflammation [43]. Mucosal Escherichia coli expressing the AFA-1 operon upregulates the expression of HIF-1α in the intestinal epithelial cells, promotes angiogenesis, and induces inflammation, leading to tumor progression [44-46].

Point 6: Figure 1: Figure labels seems to be different in figure and legends: Please make sure a, b and c in caption matches the information in figure.

Response:We have modified the Figure 1 labels to match the information.

Point 7: Line 260- “in vitro bacterial colonization’—Should it be in vivo? Please clarify.

Response: We originally meant bacteria that colonize tumors from outside the body. In lines 345 to 348, we have modified the potentially ambiguous expression in the latest version.

Point 8: Heading 6-8— For the cancer immunotherapy please include more examples where microbiome acts against the immune therapy. This would provide unbiased examples and proper information for the readers. Also, please provide more information on what type of cancer and treatments are discussed.

Response:In the latest edition we have added what we believe to be representative and appropriate examples of microbial anti-immunotherapy (refs. 162,164,165), and have refined the information on cancer types and treatments discussed.

References 162:B. Fragilis and B. Thetaiotaomicron promote tumor CTLA-4 blockade reactivity by promoting the IL-12-dependent TH1 immune response. (line 442-444)

References 164:In CRC model mice, inosine produced by the metabolism of B. Pseudolongum was confirmed to enhance anti-tumor immunity induced by anti-CTLA-4 treatment, mediated by T cells. (line 450-452)

References 165:SCFAs produced by gut microbiota inhibited CD80/CD86 upregulation on dendritic cells induced by anti-CTLA-4 treatment, limiting the efficacy of anti-CTLA-4 in mice with metastatic melanoma. (line 452-455)

Point 9: Line 300- Please provide the context of responders and non-responders.

Response:In lines 414 to 419, we clarify the definition of responder and non-responder.

Round 2

Reviewer 3 Report

Thank you for the hard work and changes you have made in the paper. Please check for following minor changes before its final version: 

1. Please check if the current version contains the updated title, it appears same to me. 

2. Rewrite line 66 and line 70 for better flow of information.